# Effect of Holder Pasteurization, Mode of Delivery, and Infant’s Gender on Fatty Acid Composition of Donor Breast Milk

**DOI:** 10.3390/nu16111689

**Published:** 2024-05-29

**Authors:** Réka Anna Vass, Miaomiao Zhang, Livia Simon Sarkadi, Márta Üveges, Judit Tormási, Eszter L. Benes, Tibor Ertl, Sandor G. Vari

**Affiliations:** 1Department of Obstetrics and Gynecology, Medical School University of Pécs, 7624 Pécs, Hungary; ertl.tibor@pte.hu; 2National Laboratory on Human Reproduction, University of Pécs, 7624 Pécs, Hungary; 3Obstetrics and Gynecology, Magyar Imre Hospital, 8400 Ajka, Hungary; 4Department of Nutrition, Faculty of Food Science and Technology, Hungarian University of Agriculture and Life Sciences, 1118 Budapest, Hungary; zhang.miaomiao@phd.uni-mate.hu (M.Z.); simonne.sarkadi.livia@uni-mate.hu (L.S.S.); 5Division of Chemical, Noise, Vibration, and Lighting Technology Laboratories, Department of Methodology and Public Health Laboratories, National Center for Public Health and Pharmacy, 1096 Budapest, Hungary; uveges.marta@uni-mate.hu; 6Department of Food Chemistry and Analysis, Faculty of Food Science and Technology, Hungarian University of Agriculture and Life Sciences, 1118 Budapest, Hungary; tormasi.judit@uni-mate.hu (J.T.); benes.eszter.luca@uni-mate.hu (E.L.B.); 7International Research and Innovation in Medicine Program, Cedars-Sinai Medical Center, Los Angeles, CA 90048, USA; sandor.vari@cshs.org

**Keywords:** capric acid, lauric acid, myristic acid, palmitic acid, stearic acid, linoleic acid, oleic acid, donor milk

## Abstract

Breast milk (BM) plays a crucial role in providing essential fatty acids (FA) and energy for the growing infant. When the mother’s own BM is not available, nutritional recommendations suggest donor milk (DM) in clinical and home practices. BM was collected from a variety of donor mothers in different lactation stages. Holder pasteurization (HoP) eliminates potential contaminants to ensure safety. FA content of BM samples from the Breast Milk Collection Center of Pécs, Hungary, were analyzed before and after HoP. HoP decreases the level of C6:0, C8:0, C14:1n-5c, C18:1n-9c, C18:3n-6c, C18:3n-3c, and C20:4n-6c in BM, while C14:0, C16:0, C18:1n-9t, C22:0, C22:1n-9c, C24:0, C24:1n-9c, and C22:6n-3c were found in elevated concentration after HoP. We did not detect time-dependent concentration changes in FAs in the first year of lactation. BM produced for girl infants contains higher C20:2n-6c levels. In the BM of mothers who delivered via cesarean section, C12:0, C15:0, C16:0, C17:0, C18:0, C18:1n-9t, C22:1n-9c levels were higher, while C18:2n-6c, C22:0, C24:0, and C22:6n-3c concentrations were lower compared to mothers who gave birth spontaneously. FAs in BM are constant during the first year of lactation. Although HoP modifies the concentration of different FAs, pasteurized DM provides essential FAs to the developing infant. Current data providing information about the FA profile of BM gives origination to supplementation guidelines.

## 1. Introduction

Human milk is a remarkable and complex fluid that has evolved over time to meet the nutritional and developmental needs of infants [1,2]. Fatty acids are crucial components of human milk and play a vital role in the growth and well-being of the child [3,4]; on average, breast milk is composed of about 3–5% fat [5]. However, it is important to note that individual variations are common. Human milk contains a variety of fatty acids that are crucial for the growth and development of infants. These fatty acids are essential in developing the nervous system, brain, and overall health [6,7]. The composition of fatty acids in human milk can vary depending on factors such as the mother’s diet [8,9], and breast milk (BM) is a concentrated source of calories, essential fatty acids, and fat-soluble vitamins. FAs synthesized de novo in the mammary gland, such as caprylic acid (8:0), capric acid (10:0), lauric acid (12:0), and myristic acid (14:0), are known as medium-chain saturated fatty acids (MCFAs) [10]. It was found that the content of MCFAs increased from colostrum to transitional and mature milk, irrespective of the region or gestational age of mothers [11,12]. The two main types of FAs found in human milk are saturated and unsaturated FAs [13].

Saturated fatty acids (SFAs): These fatty acids have no double bonds between carbon atoms. Examples include lauric acid, myristic acid, and palmitic acid. Palmitic acid is one of human milk’s most abundant saturated fatty acids [14].

Two subtypes of unsaturated fatty acids (UFAs) are distinguished: monounsaturated fatty acids (MUFAs) and polyunsaturated fatty acids (PUFAs) [14].

MUFAs are FAs that have one double bond between carbon atoms. Oleic acid is human milk’s primary monounsaturated fatty acid [15]. MUFAs have been associated with various health benefits, particularly cardiovascular health. They help improve blood lipid profiles by reducing levels of low-density lipoprotein (LDL) cholesterol while maintaining or increasing levels of high-density lipoprotein (HDL) cholesterol [16].

PUFAs have two or more double bonds between carbon atoms. The two main types of polyunsaturated fatty acids in human milk are omega-3 (e.g., docosahexaenoic acid and alpha-linolenic acid) and omega-6 (e.g., linoleic acid (LA) and arachidonic acid) [8,11]. Omega-3 fatty acids are alfa-linolenic acid (ALA), arachidonic acid (ARA), eicosatrienoic acid (ETE), eicosapentaenoic acid (EPA), and docosahexaenoic acid (DHA). DHA is a major structural component of the brain and is crucial for neurodevelopment and function. EPA and DHA have been shown to have cardiovascular benefits through triglyceride level reduction, lowering blood pressure, having anti-inflammatory effects, and impacting immune responses [17]. DHA is present in high concentrations in the retina, and adequate intake is associated with a reduced risk of age-related macular degeneration (AMD) and other disorders [18]. The placenta is crucial in nutrient transport from the mother to the fetus. Fatty acids, including essential ones like LA and ALA, are transferred across the placenta to support fetal development. In the case of preterm birth, this transfer is disrupted, therefore the only postnatal source is the BM of FAs for the developing infant [3,4,19].

Based on the number of carbon atoms in the alkyl chain length, fatty acids can be classified into short-chain (2–4 carbon atoms), medium-chain (6–10 carbon atoms), and long-chain fatty acids (12–26 carbon atoms) [20,21]. Medium-chain (saturated) fatty acids (MCFA) include caproic (6:0), caprylic (8:0), and capric (10:0) acids. Long-chain (saturated) fatty acids (LCFA) include linoleic acid (C18:2*n*-6; LA), alfa-linolenic acid (C18:3*n*-3; ALA), arachidonic acid (C20:4*n*-6; ARA), docosahexaenoic acid (C22:6*n*-3; DHA), eicosapentaenoic acid (C20:5*n*-3; EPA), behenic (22:0), lignoceric (24:0) acid [3,22].

Human breast milk also contains LCPUFAs, which play a role in the immune system. Breastfeeding protects against childhood infections [23] and may protect against childhood allergies and asthma [24], and part of this protection seems likely to be due to optimized immune development in breastfed infants. They control and participate in pain signaling pathways, inflammation, thrombosis, and vasoconstriction [25,26]. Fatty acids play a crucial role in intrauterine development, contributing to the formation and growth of various tissues and organs in the developing fetus. The availability of essential fatty acids during pregnancy is critical to developing the nervous and immune systems and other physiological processes [3,4]. Fatty acids are structural components of cell membranes, influencing membrane fluidity and stability [25]. This is particularly important during rapid cell division and differentiation in the developing fetus [3,4]. A cohort study suggests that higher mean daily serum levels of DHA during the early postnatal period are associated with less severe retinopathy of prematurity (ROP), but only in infants with elevated arachidonic acid values [27].

For infants who rely on pasteurized donor milk for various reasons, including prematurity or medical conditions or their own mother’s milk is not available, pasteurized donor milk ensures safety and nutritional support [28]. Holder pasteurization (HoP) is knowingly influencing the composition of BM [29,30,31,32,33], although when the mother’s own milk is not available this is the recommended feeding form [34]. The present work is focused on providing information about the general FA composition of BM and DM, to optimize nutritional supplementation of newborns.

## 2. Materials and Methods

### 2.1. Donor Milk Samples

After the approval of the Regional and Local Research Ethics Committee of the University of Pécs, Pécs, Hungary (PTE KK 7072-2018). Waivers for participant consent were obtained. To determine the effect of HoP on breast milk composition, 56 registered donor mothers were recruited from the Breast Milk Collection Center of the Unified Health Institution at Pécs, Hungary. Freshly pumped BM was collected according to the center’s protocol. We have chosen donations on 10 random occasions. Samples were taken individually and immediately stored at −80 °C, then the donated BM samples were pooled, and Holder pasteurized (30 min at 62.5 °C) at the Unified Health Institution (Pécs, Hungary) (Figure 1). Three samples were taken from the holder pasteurized donor milk pool and stored at −80 °C until further analysis, similar to our previous works [30,31,32,33]. Then, 3–4 mL aliquots were taken with a sterile pipette and placed into microtubes (Eppendorf, Hamburg, Germany). After labeling sample containers, they were immediately placed in a freezer at—20 °C. Samples were processed at the Department of Food Chemistry and Analysis (Budapest, Hungary).

### 2.2. Gas Chromatographic Analysis of Samples

Gas chromatographic analysis was performed to detect FAs in BM samples. The study of BM samples was conducted based on a reference ISO method [35]. The reference method was slightly modified regarding sample preparation. The frozen BM samples were thawed to room temperature and vortexed effectively to gain homogeneity, and then 0.5 g of each BM sample was transferred into conical centrifuge tubes. Following the ISO method 2.5 mL of tert-butyl methyl ether was added to the samples. During sample transesterification reactions, the reaction times were noted. First, 2.5 mL of 5% (*w*/*v*) methanolic sodium hydroxide solution was added to the tube and mixed for 10 s using a LABINCO L46 Power Mixer (Breda, The Netherlands). After a specific time (180 s), the tube was opened, and 1 mL isooctane was added to the mixture; 30 s later, the reaction was stopped when a 5 mL neutralization mixture was added. The neutralization was performed with a 10% (*w*/*v*) disodium hydrogen citrate and 15% (*w*/*v*) sodium chloride aqueous solution. Samples were centrifuged by Hettich Mikro 22R Centrifuge (Hetting Zentrifugen, Tuttlingen, Germany) for 5 min at 1750 rpm (g = 375) to obtain two different phases. The upper layer was transferred into a gas chromatography vial for GC analysis.

The resulting volatile fatty acid methyl ester compounds (FAMEs) were analyzed by an Agilent 6890 GC-FID system (GC ChemStation B.04.002 [98] (2001–2009 Agilent Technologies Inc.) Palo Alto, CA, USA), which was equipped with an autosampler (Agilent 7683), for the separation of FAMEs Phenomenex Zebron ZB-FAME (Phenomenex, Torrance, CA, USA) (60 m, 0.25 mm, 0.20 µm with cyanopropyl stationary phase) column was applied. The flow of the mobile phase (hydrogen gas) was set to 1.2 mL min^–1^. The inlet and detector temperatures were set to 250 °C and 260 °C, respectively. A 1 µL amount was injected with sample supernatants while a 50:1 split ratio was used. The gradient temperature program was as follows: 100 °C of starting oven temperature was held for 3 min, then the column was heated applying 20 °C min^–1^ rate to 166 °C (held for 5 min), then 1 °C min^–1^ gradient was used to reach 180 °C, the final temperature (240 °C) was held for 3 min that was achieved with a temperature gradient of 10 °C min^–1^. Mass Hunter software (MSD Chemstation F.01.03.2357 (1989–2015 Agilent Technologies Inc.) Palo Alto, CA, USA) was used to control GC-FID system. Chromatographic separation was carried out in 40 min per sample [36].

The investigated compounds were identified by a FAME (fatty acid methyl ester) standard mixture solution (Supelco 37 Component FAME Mix—Supleco Analytical chromatography division of Sigma Aldrich, Burlington, MA, USA) based on retention times. Results (fatty acid composition of the investigated human milk samples) were expressed as a percentage of the peak area of total fatty acid content measured in the sample.

### 2.3. Data Analysis

The Kolmogorov–Smirnov test results were considered to test the data’s normality. Multivariate data analysis was performed using principal component analysis (PCA). A support vector machine (SVM) was used to classify the datasets. SPSS Statistics 23 (IBM, New York, NY, USA) software was used to evaluate. Differences between groups were tested by Kruskal–Wallis test. Results were considered statistically significant in case p SPSS Statistics 23 (IBM, New York, NY, USA) software was used to evaluate. The Kruskal–Wallis test was performed to compare the different groups since the distribution of fatty acids did not follow the normal distribution in the analyzed samples. Subgroups were analyzed according to the mode of delivery, the infant’s gender, and different time points during lactation—1–3 vs. 3–6 vs. 6–12 months. Data were expressed as mean ± SD. Principal component analysis (PCA) was performed for data visualization and pattern recognition. The Hotelling’s T2 and F-residual values were used for outlier detection. Random seven-fold cross-validation was used to verify the model. A support vector machine (SVM) was performed to classify the data based on the fatty acid profile of the pasteurized and non-pasteurized samples. The classification was run on the scores of the samples after PCA. Linear, quadratic, and cubic kernel functions were used for the modeling. Random eleven-fold cross-validation was used to verify the model performance.

## 3. Results

### 3.1. Maternal Data

Donor mothers were generally 30.53 ± 5.44 years old at the time of donation. Their body mass index (BMI) was 24.45 ± 3.38. Infants were born between 38th and 42nd weeks of gestation, on average 39.06 ± 2.71 weeks. Of the recruited mothers, 57.6% gave birth vaginally, while in 42.4% of the cases, a cesarian section was performed. Boy infants were born in 40.6%, and girls in 59.4% of deliveries. Primiparous mothers donated 35.5% of the BM samples, while 64.5% were from multiparous mothers. None of the volunteers had chronic diseases, and every one of them took a kind of vitamin supplementation during breastfeeding. BM samples were collected in the Breast Milk Collection Center on 10 different occasions, and the pool sizes were variable from 4 to 9 samples.

### 3.2. Fatty Acids in Breast Milk

The main fatty acids of BM produced for term infants are lauric, myristic, palmitic, stearic, cis-9-octadecenoic, and methyl linoleate acids. The other 24 FAs are presented in 6.36%. The general FA composition of the analyzed raw BM samples is shown in Table 1. Of the tested fatty acids, the samples did not contain the following: C15:1n-5c; C21:0; and C20:3n-3c (Table 1).

### 3.3. Effect of Holder Pasteurization on Breast Milk Samples

Holder pasteurization changed the amount of 14 fatty acids in BM (Table 2). The concentration of caproic acid (C6:0), caprylic acid (C8:0), capric acid (C10:0), myristoleic acid (C14:1n-5c), oleic acid (C18:1n-9c), γ-linolenic acid (C18:3n-6c), α-linolenic acid (C18:3n-3c), and ARA (C20:4n-6c) decreased after HoP. In the case of myristic acid (C14:0), palmitic acid (C16:0), elaidic acid (C18:1n-9t), lignoceric acid (C24:0), nervonic acid (C24:1n-9c), and DHA (C22:6n-3c) increased concentrations were detected after HoP (Table 2).

### 3.4. Analysis of Different Co-Variants

The examined FAs showed no changes during the first 12 months of lactation. In BM produced for girl infants, C20:2n-6c presented in significantly higher concentration. When analyzing FAs based on maternal postpartum BMI, we did not find significant differences, although the donors were mainly in between the average weight BMI range. In BM samples of mothers who delivered via C-section, lower lauric acid (C12:0), pentadecylic acid (C15:0), palmitic acid (C16:0), margaric acid (C17:0), stearic acid (C18:0), elaidic acid (C18:1n-9t), and eicosenoic acid (C22:1n-9c) concentrations were determined compared to samples of mothers who delivered vaginally. Levels of linoleic acid (C18:2n-6c), behenic acid (C22:0), lignoceric acid (C24:0), and docosahexaenoic acid (C22:6n-3c) were significantly lower in the BM of mothers after spontaneous delivery (Table 3).

### 3.5. Results of Principal Component Analysis (PCA)

#### 3.5.1. Fatty Acid Profile of BM Samples

Based on the results of PCA, the first three principal components (PC) described the variance of the dataset at 98% (PC1: 60%; PC2: 28%; PC3: 10%). No outliers were found based on the values of Hotelling T2 and F-residuals. The position of the samples for the first two PCs was mainly influenced by C16:0, C17:0, C18:0, C18:1n-9c, and C18:2n-6c fatty acids. In addition, the amount of C12:0, C15:0, and C20:3n-6c fatty acids present in the samples also had a significant effect. However, according to the results obtained for the first two PCs, the samples did not separate according to the groups tested (Figure 2).

#### 3.5.2. Influence of Infant’s Gender

However, when the third PC was considered, there was a separation between the samples based on the sex of the infant. For the second and third PCs, the differences between the samples are visible. For PC2, the amount of C18:1n-9c fatty acid, while for PC3, the amount of C14:0, C12:0, and C10:0 fatty acids had a substantial effect on the position of the samples and thus on the separation. The location in the negative range for PC2 was explained by a higher amount of C18:2n-6c fatty acid, while for the samples in the positive range, it was due to C17:0. BM produced for girl infants contained higher C20:2n-6c level. The latter effect was much smaller according to the correlation loading (Figure 3).

#### 3.5.3. Effect of Holder Pasteurization

PCA was also performed on the fatty acid profiles of the pasteurized and untreated samples, the results of which are summarized in Figure 4. The first two PCs explain 93% of the variance in the data. The results clearly showed that pasteurization had a relevant effect on the fatty acid profile of the samples. This suggests that the samples, except for p10, are well separated along PC1 and PC2. Both principal components have a significant effect on the position of the samples. The pasteurized samples, except sample p1, were in the negative region of PC1, which was mainly associated with a higher percentage of C18:1n-9c fatty acid. The importance of this can be supported by the significant difference (*p* < 0.05) found for the Kruskal–Wallis test. The position in the positive region of PC1 was mainly explained by a higher proportion of C16:0 fatty acid, which was more prevalent for untreated samples (Figure 4). A classification method was also run to evaluate the effect of pasteurization. The best results were obtained with cubic support vector machines, resulting in the complete separation of samples. The accuracy of the model was 100% (Figure 4).

#### 3.5.4. Mode of Delivery

When the samples are labeled according to the mode of delivery in the PCA plot (Figure 5), they are separated based on the fatty acid profiles. In the case of PC1, this is most affected by the amount of each saturated and unsaturated fatty acid. Mothers who deliver by cesarean section have higher levels of C18:2n-6c fatty acids in their breast milk than mothers who deliver vaginally. In addition, the amounts of C20:4n-6c and C20:2n-6c fatty acids were also found to be higher. For PC2, the position of the samples is most dominated by C20:3n-6c, C12:0, and C18:1n-9c fatty acids. Samples with higher amounts of these fatty acids are in the negative region of PC1. In contrast, samples in the positive range of PC1 have a higher percentage of C16:0, C17:0, C18:0, and C15:0 fatty acids (Figure 5).

## 4. Discussion

Fatty acids are a major infant energy source, providing a dense and easily digestible form of calories. They are essential for the development of the nervous system, including the brain, and contribute to the overall growth and development of the infant [4,5]. For instance, some mothers may have milk with a higher fat content, while others may have milk with a lower fat content. The fat content is influenced by several factors, including the mother’s diet, the feeding duration, and the lactation stage. FA absorption through the gastrointestinal system is known; some FAs are synthesized by the mammary gland from carbohydrates [37].

In this study, we provide information about the general FA composition of BM and DM. The main FAs in BM are lauric, myristic, palmitic, stearic, cis-9-octadecenoic, and methyl linoleate acids. The other 24 examined FAs are present in only 6.36% [13]. This rate remains unaltered after HoP treatment.

Fatty acids in human milk also contribute to the immunological protection of the infant. Some fatty acids, such as lauric acid, have antimicrobial properties that help protect the infant from infections [38]. Breast milk provides a diverse array of fatty acids that support the development and function of the immune system [6,39]. DHA, an omega-3 fatty acid, is essential for developing the central nervous system, especially the brain. It is a major component of neuronal cell membranes and is critical for cognitive function and visual acuity. Breast milk is a significant source of DHA for infants. DHA is also crucial for developing the retina and overall visual system. Adequate levels of DHA are associated with improved visual and cognitive outcomes in infants [40,41,42,43]. Oleic acid is a monounsaturated omega-9 fatty acid, one of nature’s most common fatty acids. It is classified as a non-essential fatty acid because the human body can synthesize it, but it is also obtained through diet [44]. Oleic acid has a variety of biological functions, has an anti-inflammatory effect, and is a significant component of human milk [45,46].

Fatty acids are structural components of cell membranes. They contribute to membrane fluidity and stability, influencing cell function and signaling. This is particularly relevant in rapidly growing tissues during postnatal development [7,19]. LCPUFAs serve as a dense and easily metabolizable energy source for infants. This is essential for supporting rapid growth and providing energy demands during the early stages of life. FAs are precursors to various signaling molecules, including hormones. They play a role in hormonal regulation, essential for developing endocrine organs and overall hormonal balance [47].

Essential fatty acids, including omega-3 and omega-6 fatty acids, have multifunctional effects in preventing some diseases, such as cardiovascular, myocardial, and bowel diseases, and play a role in bone health [48]. They are involved in forming and maintaining bone tissue and can influence the balance of bone-forming and bone-resorbing cells [49]. Lauric acid has antimicrobial properties that contribute to the infant’s immune defense [50].

Breastfeeding and the unique composition of human milk have been associated with numerous long-term health benefits for infants, including a reduced risk of certain infections, allergies, and chronic diseases later in life [51,52,53]. Maternal diet influences the composition of BM [43,54] and the fetal outcome of the pregnancy [42]. In our study, the recruited mothers, based on questionnaires, followed no special diets, and all of them reportedly took vitamin supplementation.

HoP is chosen as it strikes a balance between ensuring the safety of DM and preserving as many of its nutritional components as possible. In everyday practice, no precaution is taken to prevent oxidation during pasteurization, resulting in fluctuation of FA composition of DM. Lepri et al. reported an 83% increase in FFA content after heat treatment [55]. Another study detected that after HoP, the proportion of MCFAs was increased [56]. Present result showed that HoP decreased the level of caproic (C6:0), caprylic (C8:0), cis-9-tetradecenoic (C14:1n-5c), oleic (C18:1n-9c), C18:3n-6c, alfa-linoleic (C18:3n-3c), and eicosatetraenoic (C20:4n-6c) acids in BM, while myristic (C14:0), palmitic acid (C16:0), C18:1n-9t, behenic (C22:0), C22:1n-9c, lignoceric (C24:0), C24:1n-9c, and C22:6n-3c were found in elevated concentration after HoP. Similarly to our results, in a review that analyzed over 55 publications, the authors concluded that the most abundant FAs (palmitic, linoleic, and oleic acid) remained stable over time [57]. The changes detected by our research group maintained that increase-free FAs are known to be more absorbable in the gastrointestinal system.

BM-derived components play essential roles in early life immune development. Still, exposure to antigens (e.g., from microbes and foods) and an adequate immune response are critical in shaping the infant’s gut microbiota and MCFAs, mainly 10:0, 12:0, and 14:0. These FAs stabilize the bacterial flora in the child’s digestive tract [58].

LCPUFAs in BM [43] have been identified to impact the development of the visual and cognitive systems of the infant [52]. After birth, preterm infants become dependent on external sources of LCPUFAs. Although studies indicated the potential positive effects of LCPUFA supplementation on neurodevelopment and vision, recent large, randomized trials revealed concerns about the possible increase in risk [59]. A recent meta-analysis concluded that supplementation with DHA alone increased necrotizing enterocolitis (NEC), but when ARA and DHA supplementation were combined, it increased NEC incidence [59]. LCPUFA supplementation modulated the inflammatory process in the pathophysiology of NEC [59]. Also, results indicate that enteral DHA supplementation reduced NEC incidence among preterm infants due to their immunoregulatory effect [59,60,61].

Infants who had received LCPUFAs had a lower risk of wheezing/asthma, wheezing/asthma plus atopic dermatitis, any allergy, and upper respiratory tract infection. Infants receiving LCPUFAs were less likely to develop bronchitis or bronchiolitis at 5, 7, and 9 months [38,41,62]. Also, there were fewer allergic illnesses and skin allergic illnesses in the first year compared to infants in the control group, and these outcomes remained lower at the age of 4 years. LCPUFAs reduced wheeze/asthma in infants of allergic mothers [38]. Infants with higher plasma DHA or higher plasma EPA and docosapentaenoic acid and DHA at age 6 months were less likely to develop recurrent wheezing by age 12 months [63]. In a later follow-up study, when the children were five, there was no difference between groups for any clinical outcome related to lung function or allergy [64]. A birth cohort study in Iceland identified that 2.5-year-old children who had received *n*-3 LCPUFA supplements in infancy were less likely to have been diagnosed with food sensitization [65,66]. *n*-3 LCPUFAs also decreased the severity of allergies [65]. Early *n*-3 and *n*-6 LCPUFA supply may moderate the impact of hypoxia and oxidative damage, thus affecting the recovery from injury, later organ development, and neurodevelopmental outcomes [67]. The European Food Safety Authority recommends supplementation with DHA [68].

Results indicated that the mode of delivery impacts the composition of BM FAs. In our study, similarly to previous findings [69], the BM of mothers who delivered spontaneously contained significantly higher *n*-3 PUFA, stearic, and palmitoleic acid levels, which are knowingly anti-inflammatory components. Similar to other results, in our work, the BM of mothers undergoing C-sections had elevated DHA BM levels [69]. Our results showed that inflammatory FA content and *n*-6/*n*-3 ratio were higher in the BM of mothers who delivered via C-section.

Previous studies reported significant sex-based differences in the FA composition of BM; fat content produced for boy infants was significantly higher than for girl infants [70]. A higher ratio of ARA/DHA in female infants may reduce the risk of food sensitization and atopic dermatitis [71]. We found that BM produced for female infants contains higher eicosadienoic acid levels. Low temperature treatments, like Holder pasteurization, create oxidative environments, resulting in modest changes in FA concentration [72].

Our results give information about the fatty acid profile of BM, supporting the aims of supplementation among preterm infants. Although HoP resulted in changes in the detected proportion of FAs in BM, due to the statistically significant but infinitesimal modification, pasteurized DM provides an alternative to own mother’s milk during postnatal adaptation. Pooled DM is a safe way to merge BM from different lactation sessions and to evenly distribute nutrients. Different treatment technologies are applied to ensure the microbiological safety of DM, like high-temperature, short-time (HTST), or ultra-violet processing. Previous works investigating these techniques had no significant effect on the FA composition of BM [73,74]. Serum hormone levels in preterm infants, e.g., leptin, predict early childhood developmental assessment scores [75]. Studies proving the ability of fatty acids to improve cognitive functions among adults are known. However, the possible predictive effects of fatty acids are still unknown despite their multifunctional effect on neurodevelopment [76,77]. Fatty acid supplementation for preterm infants is a common practice in neonatal care to support their growth and development [78], and it is especially relevant because these infants are born before they have had the chance to accumulate sufficient fat stores in utero.

## 5. Conclusions

The adaptability of human milk to the child’s specific needs is a fascinating aspect of breastfeeding and highlights the evolutionary importance of this unique source of nourishment. The FA profile of human milk is a critical aspect of its nutritional and immunological properties, contributing to the overall health and well-being of the breastfed infant. Based on our results, the fatty acid composition of BM differs based on the gender of the infants, the course of delivery, the maternal age, and BMI. While the decrease in fatty acids is a concern, the reduction in nutritional components, including PUFAs, needs to be balanced against the primary goal of pasteurization, which is to destroy harmful microorganisms while minimizing the impact on milk quality. Holder pasteurization has been widely used because it effectively reduces microbial contamination while, as our measurements strengthen previous results, minimizing the impact on the overall composition of breast milk. Data from the present study provides detailed information about the FA profile of BM; therefore, it may contribute to protocols for FA supplementation of premature or formula-fed infants. Fatty acid supplementation is one component of the comprehensive care approach, which provides optimal nutrition with the supplementation of fatty acids and supports development and growth.

## Figures and Tables

**Figure 1 nutrients-16-01689-f001:**
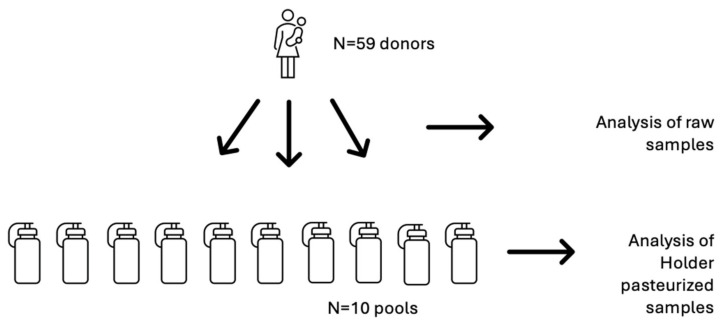
Schematic figure of sample collection and analysis.

**Figure 2 nutrients-16-01689-f002:**
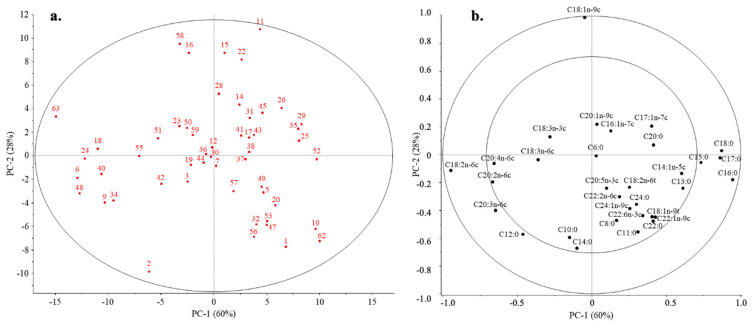
PCA scores (**a**) and correlation loadings (**b**) obtained for all of the investigated BM samples based on their fatty acid profile.

**Figure 3 nutrients-16-01689-f003:**
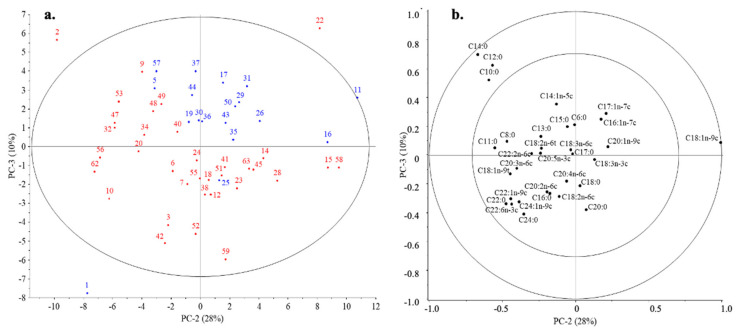
Influence of infant’s gender on the separation (PC2–PC3) of BM samples according to the fatty acid profile. On scores plot (**a**) the samples were marked with colors according to the sex of the infant: girl—red; boy—blue. Fatty acids influence the position of the samples presented on the correlation loadings plot (**b**).

**Figure 4 nutrients-16-01689-f004:**
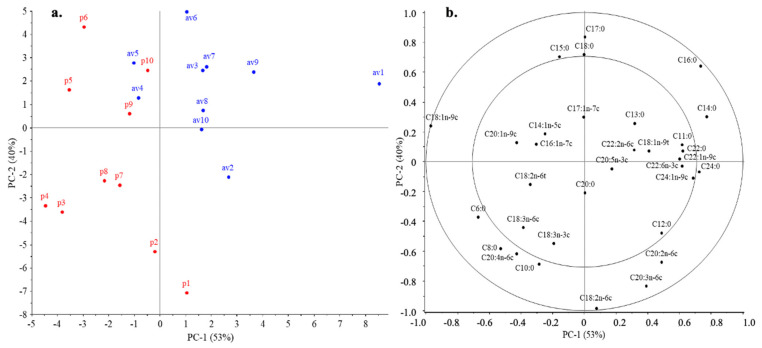
Separation of Holder pasteurized pooled BM samples (indicated with red color) and raw BM samples (indicated with blue color) based on the fatty acid profiles can be seen on scores plot (**a**). Correlation loadings (**b**) present fatty acids and their effect on the position of the scores.

**Figure 5 nutrients-16-01689-f005:**
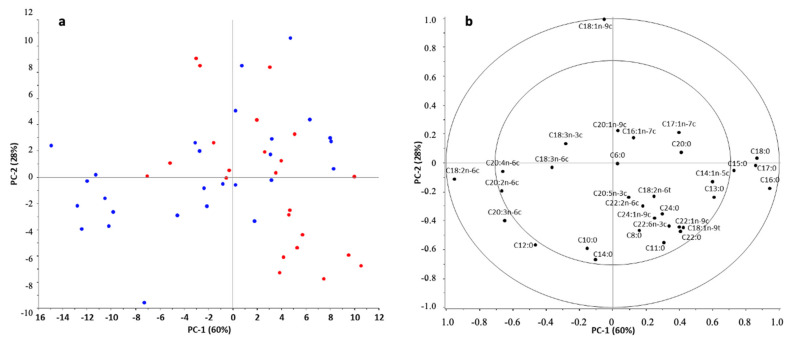
Effect of mode of delivery on the PCA (scores (**a**) and correlation loadings (**b**) of PC1-PC-2) results based on the fatty acids measured in breast milk samples. BMs provided by mothers with cesarean sections are labeled with blue dots, while samples provided by mothers with spontaneous delivery are colored with red dots.

**Table 1 nutrients-16-01689-t001:** Fatty acid composition of the analyzed breast milk samples (% of total fatty acid profile in mean ± STD).

Fatty Acids	Common Name	Value (%)	Fatty Acids	Common Name	Value (%)
C6:0	Caproic acid	0.01 ± 0.01	C18:2n-6t	Linolelaidic acid	<0.01
C8:0	Caprylic acid	0.06 ± 0.03	C18:2n-6c	Linoleic acid (LA)	15.13 ± 5.02
C10:0	Capric acid	0.98 ± 0.20	C18:3n-6c	γ-Linolenic acid (GLA)	0.10 ± 0.06
C11:0	Undecanoic acid	<0.01	C18:3n-3c	α-Linolenic acid (ALA)	0.54 ± 0.18
C12:0	Lauric acid	5.46 ± 1.92	C20:0	Arachidic acid	0.18 ± 0.08
C13:0	Tridecylic acid	0.03 ± 0.02	C20:1n-9c	Eicosenoic acid	0.40 ± 0.11
C14:0	Myristic acid	7.02 ± 1.99	C20:2n-6c	Eicosadienoic acid	0.30 ± 0.09
C14:1n-5c	Myristoleic acid	0.14 ± 0.06	C20:3n-6c	Eicosatrienoic acid (ETE)	0.35 ± 0.08
C15:0	Pentadecylic acid	0.31 ± 0.15	C20:4n-6c	Arachidonic acid (ARA)	0.40 ± 0.08
C16:0	Palmitic acid	26.01 ± 4.38	C22:0	Behenic acid	0.01 ± 0.03
C16:1n-7c	Palmitoleic acid	1.90 ± 0.44	C22:1n-9c	Erucic acid	0.02 ± 0.03
C17:1n-7c	Heptadecenoic acid	0.12 ± 0.10	C20:5n-3c	Eicosapentaenoic acid (EPA)	<0.01
C17:0	Margaric acid	0.27 ± 0.09	C22:2n-6c	Docosadienoic acid	0.01 ± 0.02
C18:0	Stearic acid	7.73 ± 1.87	C24:0	Lignoceric acid	0.02 ± 0.05
C18:1n-9t	Elaidic acid	0.12 ± 0.27	C24:1n-9c	Nervonic acid	0.02 ± 0.08
C18:1n-9c	Oleic acid	32.29 ± 4.06	C22:6n-3c	Docosahexaenoic acid (DHA)	0.07 ± 0.16

**Table 2 nutrients-16-01689-t002:** Effect of Holder pasteurization on breast milk (% of total fatty acid profile in mean ± STD).

Fatty Acid	Raw	S	Pasteurized	Fatty Acid	Raw	S	Pasteurized
C6:0	0.07 ± 0.03	*	0.01 ± 0.00	C18:2n-6t	<0.01		<0.01
C8:0	0.19 ± 0.04	*	0.06 ± 0.02	C18:2n-6c	17.10 ± 2.47		15.21 ± 1.63
C10:0	1.26 ± 0.14	*	1.00 ± 0.07	C18:3n-6c	0.15 ± 0.03	*	0.10 ± 0.03
C11:0	<0.01		<0.01	C18:3n-3c	0.65 ± 0.12	*	0.54 ± 0.07
C12:0	5.88 ± 0.83		5.55 ± 0.65	C20:0	0.19 ± 0.02		0.18 ± 0.02
C13:0	<0.01		0.03 ± 0.01	C20:1n-9c	0.38 ± 0.05		0.39 ± 0.06
C14:0	6.05 ± 0.89	*	7.03 ± 0.68	C20:2n-6c	0.31 ± 0.05		0.30 ± 0.05
C14:1n-5c	0.16 ± 0.02	*	0.14 ± 0.02	C20:3n-6c	0.39 ± 0.07		0.36 ± 0.05
C15:0	0.25 ± 0.04		0.29 ± 0.06	C20:4n-6c	0.51 ± 0.06	*	0.40 ± 0.05
C16:0	22.98 ± 2.06	*	26.10 ± 1.16	C22:0	<0.01		0.014 ± 0.015
C16:1n-7c	1.94 ± 0.18		1.90 ± 0.18	C22:1n-9c	<0.01		0.02 ± 0.02
C17:0	0.24 ± 0.02		0.27 ± 0.03	C20:5n-3c	<0.01		<0.01
C17:1n-7c	0.08 ± 0.09		0.11 ± 0.03	C22:2n-6c	<0.01		<0.01
C18:0	6.98 ± 0.75		7.71 ± 0.61	C24:0	0.01 ± 0.03	*	0.02 ± 0.03
C18:1n-9t	0.01 ± 0.03	*	0.11 ± 0.13	C24:1n-9c	0.02 ± 0.05	*	0.03 ± 0.05
C18:1n-9c	34.16 ± 3.59	*	32.05 ± 2.46	C22:6n-3c	0.02 ± 0.07	*	0.07 ± 0.09

S = significance * *p* < 0.05.

**Table 3 nutrients-16-01689-t003:** Fatty acid content of BM in case of mothers who delivered spontaneously or via C-section (% of total fatty acid profile in mean ± STD).

Fatty Acid	Spontaneously	S	C-Section	Fatty Acid	Spontaneously	S	C-Section
C6:0	0.01 ± 0.01		0.01 ± 0.01	C18:2n-6t	<0.01		<0.01
C8:0	0.06 ± 0.03		0.06 ± 0.03	C18:2n-6c	14.59 ± 4.84	*	15.25 ± 4.74
C10:0	1.01 ± 0.19		0.95 ± 0.21	C18:3n-6c	0.08 ± 0.06		0.11 ± 0.06
C11:0	<0.01		<0.01	C18:3n-3c	0.53 ± 0.16		0.54 ± 0.20
C12:0	5.60 ± 1.90	*	5.19 ± 1.98	C20:0	0.17 ± 0.10		0.20 ± 0.06
C13:0	0.03 ± 0.02		0.03 ± 0.02	C20:1n-9c	0.40 ± 0.11		0.40 ± 0.11
C14:0	7.19 ± 1.80		6.83 ± 2.22	C20:2n-6c	0.29 ± 0.11		0.30 ± 0.08
C14:1n-5c	0.14 ± 0.06		0.14 ± 0.06	C20:3n-6c	0.34 ± 0.08		0.36 ± 0.08
C15:0	0.31 ± 0.16	*	0.29 ± 0.14	C20:4n-6c	0.38 ± 0.07		0.42 ± 0.08
C16:0	26.57 ± 4.53	*	26.19 ± 4.37	C22:0	0.01 ± 0.03	*	0.02 ± 0.03
C16:1n-7c	1.88 ± 0.37		1.92 ± 0.51	C22:1n-9c	0.02 ± 0.03	*	0.02 ± 0.04
C17:0	0.29 ± 0.09	*	0.26 ± 0.08	C20:5n-3c	0.01 ± 0.03		<0.01
C17:1n-7c	0.10 ± 0.10		0.14 ± 0.09	C22:2n-6c	<0.01		0.01 ± 0.02
C18:0	8.15 ± 1.96	*	7.45 ± 1.57	C24:0	0.02 ± 0.06	*	0.03 ± 0.07
C18:1n-9t	0.14 ± 0.31	*	0.11 ± 0.22	C24:1n-9c	0.03 ± 0.10		0.03 ± 0.11
C18:1n-9c	31.58 ± 4.33		32.66 ± 4.03	C22:6n-3c	0.07 ± 0.17	*	0.09 ± 0.18

S = significance * *p* < 0.05.

## Data Availability

The data is not publicly available due to ethical reasons, on request it is available from the corresponding author.

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
