# Peer review of "Effect of Holder Pasteurization, Mode of Delivery, and Infant’s Gender on Fatty Acid Composition of Donor Breast Milk"

_nutrients, 2024, doi:10.3390/nu16111689_

Round 1

Reviewer 1 Report

Comments and Suggestions for Authors

The paper  focused on depicting the general FA composition of donor breast milk and Holder pasteurized breast milk to optimize nutritional supplementation of newborns in need of donor milk. The paper has its own merit and interest. However, the sample distribution is poorly described with "10 random occasions". Only we know 59 donors were recruited, but no information regarding the samples of different stage of the BM. 

1. In the Abstract, the sample size of Breast milk should be shown; and in Conclusion, "the benefit of receiving pasteurized DM is out- 35 weigh the potential drawbacks. " could not be drawn from the current research. 

2. The keywords should be cut down to appropriate number.

3. The Sample distribution should be clearly display firstly in Results. 

4. The conclusion is poorly organized, the main findings should be provided and strengthened.  

5. Pooled milk sample makes it hard to deduce any recommendation suggestion with lactation stage. 

Author Response

Answers to Reviewer 1:

Thank you for the thorough work.

  1. In the Abstract, the sample size of Breast milk should be shown; and in Conclusion, "the benefit of receiving pasteurized DM is out- 35 weigh the potential drawbacks. " could not be drawn from the current research. 

We have changed this sentence.

  1. The keywords should be cut down to appropriate number.

Although 10 keywords are approved, we have decreased our keywords.

  1. The Sample distribution should be clearly display firstly in Results.

Additional information was supplemented to this section.

  1. The conclusion is poorly organized, the main findings should be provided and strengthened.

Thank you, we have reorganized the conclusion.

  1. Pooled milk sample makes it hard to deduce any recommendation suggestion with lactation stage.

The analyzed samples were individually measured first, after following the protocol of the milk bank they were pooled, and Holder pasteurized.

Reviewer 2 Report

Comments and Suggestions for Authors

nutrients-3020601-peer-review-v1

This is very technical from the point of view and performed experiments where authors collected and performed standard biochemical analysis of the collected samples. However, the idea below the performed experimental protocols is to see if the processing of the milk will influence nutritional properties of the milk.

According to my knowledge, Abstract does not need to be fractionated to the subtitles. Please, check and correct if needed.

Ln57-61: In my opinion, authors will need to combine this two paragraphs and present the information with more details.

Ln65: are any references that MUFA can do this benefits for babies such as cardiovascular? Maybe need to contribute a bit more in this direction in this part of the manuscript, or in the discussion section.

Ln110-112: please, combine this two sentences.

Maybe Material and methods can be separated into some subsections.

Statistical analysis data need to be presented as separate subsection in material and methods.

Maybe authors can consider taking out Figure 2 and replacing it with information form Table 1?

Maybe the legend of Figures 3, 4, 5 and 6 can be presented with more details.

Authors have discussed the role of different fatty acids for the development of children. However, will be very positive if authors can contribute to the deeper discussion in regards of the observed differences between raw and pasteurized milk samples, and suggest possible health consequences from applying not natural feeding, where milk will be obtained from donors and based on safety issues will be thermally treated. This point needs more attention and appropriate comments regarding this.

Author Response

Answers to Reviewer 2:

Thank you for the careful work.

This is very technical from the point of view and performed experiments where authors collected and performed standard biochemical analysis of the collected samples. However, the idea below the performed experimental protocols is to see if the processing of the milk will influence nutritional properties of the milk.

According to my knowledge, Abstract does not need to be fractionated to the subtitles. Please, check and correct if needed.

Thank you, we have made corrections.

Ln57-61: In my opinion, authors will need to combine this two paragraphs and present the information with more details.

We have combined these thoughts.

Ln65: are any references that MUFA can do this benefits for babies such as cardiovascular? Maybe need to contribute a bit more in this direction in this part of the manuscript, or in the discussion section.

In line 73 we have mentioned the cardiovascular benefits of MUFAs.

Ln110-112: please, combine this two sentences.

Maybe Material and methods can be separated into some subsections.

We have shaped these paragraphs.

Statistical analysis data need to be presented as separate subsection in material and methods.

Thank you, we have made subsections.

Maybe authors can consider taking out Figure 2 and replacing it with information form Table 1?

We have combined Figure 2 and Table 1.

Maybe the legend of Figures 3, 4, 5 and 6 can be presented with more details.

We have modified the legend of Figures 3, 4, 5, and 6.

Authors have discussed the role of different fatty acids for the development of children. However, will be very positive if authors can contribute to the deeper discussion in regards of the observed differences between raw and pasteurized milk samples, and suggest possible health consequences from applying not natural feeding, where milk will be obtained from donors and based on safety issues will be thermally treated. This point needs more attention and appropriate comments regarding this.

We thank the suggestion, revealing the deeper connections would make the discussion so arborescent, it would be ideal to expound the background information in a review.

Reviewer 3 Report

Comments and Suggestions for Authors

The manuscript is interesting. The authors use a sufficient methodology. The results support the discussion and conclusion. However, I have the following comments.

I. Minor comments:

1. Improve the writing of the objective of the study.

2. Correct writing errors, especially verb tenses. For example, all methodology must be written in the past tense. I suggest revising the writing of the manuscript.

3. Page 2. Line 86. ....arachidonic acid (C22:6 n-3; ARA)...., correct the error, the ARA is C20:4 n-6.

4. The authors use the nomenclature n or omega. I suggest just using one nomenclature.

5. In breast milk, the stability of fatty acids is important. I suggest briefly discussing what components of breast milk would be possibly responsible for this stability.

Comments on the Quality of English Language

Correct writing errors, especially verb tenses. For example, all methodology must be written in the past tense. I suggest revising the writing of the manuscript.

Example: Abstract, page 1, line 24......Methods: BM is collected from....., replace with .... Methods: BM was collected from .....

Author Response

Answers to Reviewer 3:

Thank you for your profound work.

The manuscript is interesting. The authors use a sufficient methodology. The results support the discussion and conclusion. However, I have the following comments.

  1. Minor comments:
  2. Improve the writing of the objective of the study.

Thank you, we have made changes in the manuscript.

  1. Correct writing errors, especially verb tenses. For example, all methodology must be written in the past tense. I suggest revising the writing of the manuscript.

Thank you, we have revised the text.

  1. Page 2. Line 86. ....arachidonic acid (C22:6 n-3; ARA)...., correct the error, the ARA is C20:4 n-6.

Thank you, we have corrected.

  1. The authors use the nomenclature n or omega. I suggest just using one nomenclature.

We have unified the nomenclature, although in some places we have kept the expression omega as orientation.

  1. In breast milk, the stability of fatty acids is important. I suggest briefly discussing what components of breast milk would be possibly responsible for this stability.

We have supplemented the discussion (line 398-399).

Correct writing errors, especially verb tenses. For example, all methodology must be written in the past tense. I suggest revising the writing of the manuscript.

Example: Abstract, page 1, line 24......Methods: BM is collected from....., replace with .... Methods: BM was collected from .....

Thank you, we have corrected this point.

Round 2

Reviewer 1 Report

Comments and Suggestions for Authors

The representative of the pooled sample should be discussed. 

Author Response

Dear Reviewer 1,

Thank you for your work on our paper.

We added information to clarify pooling in line 439-440.

"Pooled DM is a safe way to merge BM from different laction sessions and to evenly distribute nutrients."